# Development of a nomogram for overall survival prediction in primary upper lobe lung cancer patients: A SEER population-based analysis

**Wenze Yu, Lu Long, Qizhuo Hou, Bin Yi***

Department of Clinical Laboratory, Xiangya Hospital, Central South University, Changsha, Hunan, China

* xyyibin@163.com

## Abstract

### Background

The upper lobe is the most common site of primary lung cancer, however, very few reports focus on its prognosis. This study aims to identify prognostic factors of lung cancer in the upper lobe, as well as to establish an effective nomogram for individualized overall survival (OS) prediction.

### Methods

Patients diagnosed with lung cancer were collected from the Surveillance, Epidemiology, and End Results Program (SEER) database for the period of 2010–2017,as recorder in the 2021 SEER database release. The demographic characteristics and OS differed in the primary sites of the upper, middle and lower lobes were drawn. The primary upper lobe lung cancer patients were further stratified by the risk indicators including Mets at DX-bone, stage, histology, grade and sex; and their OS differences in stratification were compared by the Kaplan-Meier method and the Log-Rank test. The univariate and the multivariate Cox regression were employed to determine the independent prognostic factors for the primary upper lobe lung cancer and to build a nomogram model for its OS prediction.

### Results

Depending on the different primary sites of lung cancer occurrence, all the collected patients were divided into three groups of the upper lobe (30295 individuals), the middle lobe (2801 individuals) and the lower lobe (16757 individuals), where the upper lobe group gained our attention with the largest population and an overwhelmingly low OS compared to the middle lobe group (*P <0.0001*). With the results of the univariate and multivariate Cox regression model analyses, age, sex, grade, histology type, stage, regional lymph nodes removed, bone metastasis and liver metastasis were selected as the prognostic factors and a prediction nomogram model was built. The calibration curves showed no significant bias from the reference line and the concordance index between the survival

**Data availability statement:** All data can be obtained from the Surveillance, Epidemiology, and End Results (SEER) Program website (www.seer.cancer.gov). Registered researchers can download these data for free (https://seer.cancer.gov/data). Others will be able to access or request these data in the same way as the authors. The authors confirm no special access privileges were used

**Funding:** This work was supported by the Natural Science Foundation of Hunan Province, China (Grant Numbers 2023JJ40971 and 2023JJ30965). The funders had no role in study design, data collection and analysis, decision to publish, or preparation of the manuscript.

**Competing interests:** The authors have declared that no competing interests exist.

nomogram prediction and the actual outcome for 2-year and 3-year OS was 0.761 (95% CI, 0.757–0.765). The time-dependent receiver operating characteristic curves showed that the areas under curve for 2-year and 3-year OS were 0.840 and 0.836, respectively.

## Conclusion

A novel nomogram was established which achieved good performance in predicting the probability of OS in the primary upper lobe lung cancer, indicating its potential value in individualized prediction of the clinical outcome in these patients.

## Introduction

Lung cancer is one of the most common malignancies with the highest incidence and mortality rates globally [1]. In 2022, over 2.4 million new cases of lung cancer were diagnosed worldwide, with approximately 1.8 million deaths, and the five-year survival rate remains as low as 20% [2]. In China, lung cancer accounts for 28.4% of all cancer-related deaths, making it the leading cause of cancer mortality [3]. The substantial social and economic burden underscores the urgency of addressing this public health crisis [4].

Early detection and early treatment are key strategies to improve the survival rates and quality of life for patients with lung cancer. The application of imaging techniques such as low-dose computed tomography (LDCT) has significantly enhanced the efficiency of early diagnosis [5,6], while advancements in molecular targeted therapy and immunotherapy have provided new treatment options for patients with advanced disease. However, due to the heterogeneity of the disease and the inaccuracy of prognostic prediction, some patients still do not receive appropriate treatment [7,8].

In a retrospective study by Nilssen et al. (2024), primary upper lobe lung cancer accounted for 62.3% of lung cancer cases.Its anatomical adjacency to the mediastinal lymphatics and vasculature predisposes the tumor to early nodal metastasis, with a higher mediastinal involvement rate compared to lower lobe tumors [9].Despite these findings, no prognostic tools address the unique biology of upper lobe lung cancer, relying instead on generic TNM staging.

We analyzed 15,342 primary upper lobe cases from the SEER database (2010–2020) to develop the first dedicated nomogram predicting 2- and 3-year survival. By synthesizing clinical, therapeutic, and pathological variables, this model empowers risk-adapted strategies such as neoadjuvant immunotherapy escalation or lymphadenectomy optimization, advancing personalized management of upper lobe lung cancer.

## Materials and methods

### Data resource and patient selection

Demographics, clinical characteristics and prognostic outcomes of the lung cancer patients were obtained from the 2021 SEER Program, which provides cancer statistics among the U.S. population. Cases from 2010 to 2017 were extracted and individual cancer records were generated from SEER*Stat 8.3.9.2 software. Meanwhile, patients were excluded if: (1) age at diagnosis below 20 years; (2) survival time was recorded as 0; (3) data on age, race, sex, primary site, grade (thru 2017), histology type, SEER Combined Summary Stage 2000 (2004–2017), regional lymph nodes removed, SEER Combined Mets at DX-bone (2010+) and SEER Combined Mets at DX-liver (2010+) were unavailable; or (4) other primary sites except for upper, middle and lower lobes.

## Variable extraction and outcome definition

Several variables were extracted from the SEER, including age at diagnosis, race, sex, year of diagnosis, grade, histology type, stage, regional lymph nodes removed, Mets at diagnosis-Bone(Mets at DX-Bone), Mets at diagnosis-Liver(Mets at DX-Liver), survival months and vital status. Patient outcome was described by OS, defined as the length of time from the date of diagnosis to the date of death for any reason. In this study, variables were conformed to the SEER Program Coding and Staging Manual 2021. The primary sites were classified as upper lobe (C34.1), middle lobe (C34.2) and lower lobe (C34.3). Cancers were graded as Grade I, well differentiated; Grade II, moderately differentiated; Grade III, poorly differentiated; and Grade IV, undifferentiated or anaplastic. Histology types fell into epithelial neoplasms (8010–8049), squamous cell neoplasms (8050–8089), adenomas and adenocarcinomas (8140–8389) and others. Stages included regional lymph nodes involved only, regional by direct extension only, regional by both direct extension and lymph node involvement, distant site(s) involved and localized only. Metastasis included Mets at DX-Bone (Yes/No) and Mets at DX-Liver (Yes/No). Other clinical variables included regional lymph nodes removed (Yes/No). Based on the criteria above, a total of 49853 lung cancer cases were enrolled.

## Statistical analysis

Overall survival of each subgroup was analyzed by the Kaplan-Meier method, and the statistical difference was calculated with the log-rank test. The Cox proportional hazard regression model was used for univariate or multivariate survival assessment. The multivariate Cox proportional model started with the backward stepwise selection and the Akaike information criterion (AIC) to identify the independent prognostic factors. Meanwhile, a 95% confidence interval (CI) was presented, and a forest plot for hazard ratios (HR) was made from the multivariate Cox analysis.

The nomogram was constructed using the 'rms' package (version 6.7–1) in R software, which automatically scales the regression coefficients (β) from the final multivariate Cox model to assign weighted points to each variable. This scaling process converts the β values to a 0–100 point system, where variables with larger absolute β values (indicating stronger prognostic impact) receive proportionally higher points. The total points from all variables are then mapped to the predicted survival probabilities on the nomogram's bottom scale, as per the package's default algorithm [10,11].

Based on the variables selected, a nomogram was established to predict the probability of 2-year and 3-year OS. The concordance index, receiver operating characteristic (ROC) and calibration curves were performed for nomogram evaluation. The concordance index was adopted to assess the performance between the nomogram prediction and the actual outcomes. The closer to 1.0, the better the concordance is. In our study, the time-dependent area under the receiver operating characteristic (AUC) curve was carried out to evaluate the discriminative ability of the nomogram, where the AUC of 0.7 or above indicates good separation from other outcomes. The calibration curve was modeled to determine the relationship between the nomogram-predicted survival and the actual outcomes. In a perfect calibration model, the predicted line should fall on a 45-degree diagonal line. Bootstrapping with 1000 resamples was used for internal validation and overfitting bias mitigation.

In addition, the R packages including 'survminer', 'foreign', 'survival', 'forestplot', 'rms' and 'survivalROC' were incorporated into the Kaplan–Meier analysis curves, the Cox proportional hazards regression models, the forest plots, the nomograms, and the ROC and calibration curves. A two-sided $P < 0.05$ represented the statistical significance. All statistical analyses were conducted using R software (version 4.1.2) and RStudio software (version 2021.09.1–372).

## Results

### Incidence and survival analysis

According to the criteria of inclusion and exclusion, lung cancer patients registered in the SEER Program from 2010 to 2017 were enrolled. Among them, 30295 individuals were identified with the primary site in the upper lobe, 2801 in the middle lobe, and 16757 in the lower lobe, demonstrating that the upper lobe was the most frequently occurred primary site of lung cancer (Fig 1A). Data analysis of the annual number of the primary upper lobe lung cancer patients from 2010 to 2017 revealed that as age increased, the number of the patients grew year by year, with its peak at 70–74 years old, and then decreased (Fig 1B). Compared to the OS in the middle lobe group, the OS in the upper lobe group was significantly low with a median of 25 months (*P < 0.0001*), as depicted in Fig 2.

### Demographic characteristics of the study population

A total of 30295 lung cancer patients with the primary site at the upper lobe were included. Of them, 27707 patients (91.5%) were ≥ 55 years old at diagnosis, 24213 were white patients (79.9%) and 15818 were males (52.2%); moreover, Grade III was the most frequent tumor grade (47.8%), followed by Grade II (35.3%), I (12.1%) and IV (4.9%), as illustrated in Table 1. Adenomas and adenocarcinomas prevailed in the histology types (54.2%), followed by squamous cell neoplasms (31.3%) and epithelial neoplasms (12.8%). Distant site(s) involved was the major stage (36.2%), followed by localized only (32.3%), regional by direct extension only (13.3%), regional lymph nodes involved only (9.2%) and regional by both direct extension and lymph node involvement (9.0%). 10.9% of the patients had Mets at DX-bone and 5.2%, Mets at DX-liver. Regarding the treatment schemes, 44.6% of the patients were removed the regional lymph nodes.

### OS analysis using Kaplan–Meier survival curve

We used the Kaplan-Meier method to compare the OS in patients with primary lung cancer in the upper lobe (Fig 3), who were further stratified by risk indicators excavated by data mining in Table 1. The Log-Rank tests for all Kaplan-Meier survival curves were statistically

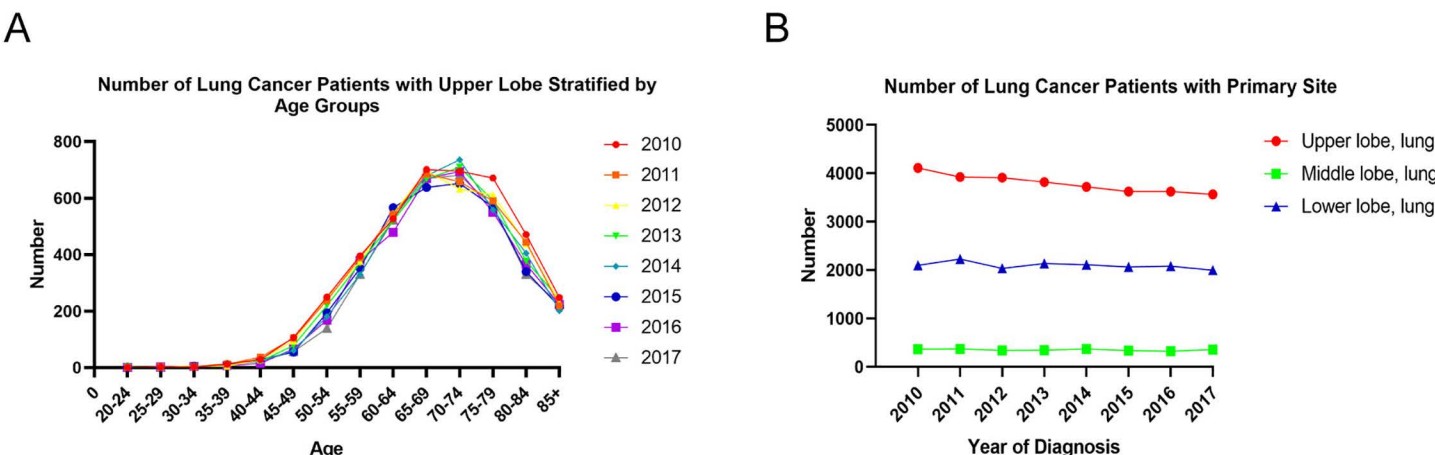

**Fig 1. Demographic characteristics of the lung cancer patients differed in primary sites.** (A) Numbers of the lung cancer patients differed in the primary sites of the upper, middle and lower lobes; (B) Numbers of the primary upper lobe lung cancer patients differed in age at diagnosis.

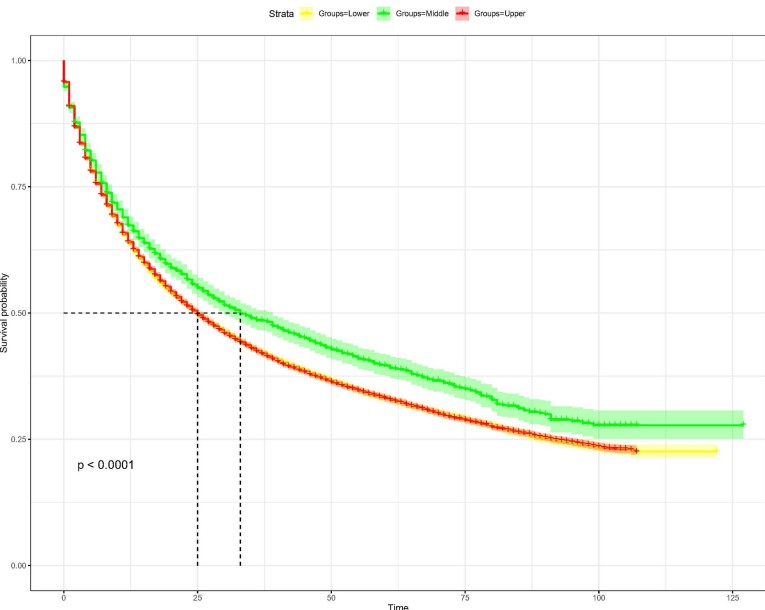

**Fig 2. Overall survival curves of the lung cancer patients differed in primary sites of the upper, middle and lower lobes.**

significant (*P <0.05*). As shown in Fig 3A, patients with Mets at DX-bone had a worse OS rate of 14% at 20 months (*P <0.0001*). Unexpectedly, patients with regional by both direct extension and lymph node involvement had the worst OS (*P <0.0001*, Fig 3B). In terms of histology type, patients with the epithelial neoplasms were significantly correlated to all-cause death, followed by squamous cell neoplasms (*P <0.0001*, Fig 3C). The male patients had a shorter OS than the female patients (*P< 0.0001*, Fig 3E). Interestingly, we found that some factors, including Mets at DX-bone, regional by both direct extension and lymph node involvement, epithelial neoplasms and male, were significantly associated with the worst clinical outcome (Fig 3). Additionally, Fig 3D depicted that the Grade Ⅳ patients were strongly correlated to a poor OS.

## Prognostic factors for OS prediction

Demographic and characteristic factors of clinical importance were selected as candidate variables for OS prediction. Ten variables were included in the univariate Cox analysis. The results showed that age, race, sex, grade, histology type, stage, regional lymph nodes removed, Mets at DX-bone and Mets at DX-liver were identified as OS-related variables (Table 2). Afterwards, the multivariate Cox analysis was performed, revealing that older age, male, higher grade, histology type of epithelial neoplasms, stage, without regional lymph nodes removed, Mets at DX-bone and Mets at DX-liver were independently associated with a poor OS in lung cancer patients with primary site in the upper lobe (Table 2). The results of the multivariate Cox analysis were intuitively displayed in the forest plot in Fig 4.

## Development and validation of the prognostic nomogram

Based on the multivariable Cox analysis, all the aforementioned variables that showed significant predictive power were incorporated into the development of the nomogram. Consequently, factors such as age, sex, grade, histology type, stage, whether regional lymph nodes were removed, and

**Table 1. Demographics and clinical characteristics of the lung cancer patients with primary site in the upper lobe.**

| Patient characteristics | Category | Number | Percentage (%) |
|---|---|---|---|
| Age at diagnosis | 20-54 | 2588 | 8.5 |
| | 55-74 | 18016 | 59.5 |
| | ≥75 | 9691 | 32.0 |
| Race | White | 24213 | 79.9 |
| | Black | 3545 | 11.7 |
| | Other | 2537 | 8.4 |
| Sex | Male | 15818 | 52.2 |
| | Female | 14477 | 47.8 |
| Year of diagnosis | 2010-2013 | 15765 | 52.0 |
| | 2014-2017 | 14530 | 48.0 |
| Grade | I | 3655 | 12.1 |
| | II | 10684 | 35.3 |
| | III | 14478 | 47.8 |
| | IV | 1478 | 4.9 |
| Histology type | Epithelial neoplasms | 3887 | 12.8 |
| | Squamous cell neoplasms | 9467 | 31.3 |
| | Adenomas and adenocarcinomas | 16419 | 54.2 |
| | Other | 522 | 1.7 |
| Stage | Regional lymph nodes involved only | 2779 | 9.2 |
| | Regional by direct extension only | 4024 | 13.3 |
| | Regional by both direct extension and lymph node involvement | 2723 | 9.0 |
| | Distant site(s) involved | 10980 | 36.2 |
| | Localized only | 9789 | 32.3 |
| Regional lymph nodes removed | Yes | 13509 | 44.6 |
| | No | 16786 | 55.4 |
| Mets at DX-Bone | Yes | 3302 | 10.9 |
| | No | 26993 | 89.1 |
| Mets at DX-Liver | Yes | 1574 | 5.2 |
| | No | 28721 | 94.8 |

presence of metastasis at diagnosis in bone (Mets at DX-bone) and liver (Mets at DX-liver) were included to construct a nomogram for predicting the prognosis of patients with primary upper lobe lung cancer. Fig 5 illustrates an example of using the nomogram to predict the survival probability of a patient. The patient is a 50-year-old male diagnosed with stage I squamous cell carcinoma of the lung who underwent lymph node dissection. The contribution of each variable to the nomogram is weighted according to its regression coefficient. For an individual patient, black dots are placed on each variable axis. A red line is drawn upward from these points to determine the score for each variable. The total score (89) is plotted on the total points axis, and a downward line is drawn to the survival axis to estimate the 2-year (76%) and 3-year (70%) OS probabilities.

The concordance index（C-index）between the survival nomogram prediction and the actual outcome for 2-year and 3-year OS was 0.761 (95% CI, 0.757–0.765). Besides, the time-dependent ROC curves showed that the AUC in 2-year and 3-year were 0.840 and 0.836, respectively (Fig 6). Both the concordance index and the AUC indicated good prediction performance of the nomogram. Fig 7 presents the calibration curves for 2-year (A) and 3-year

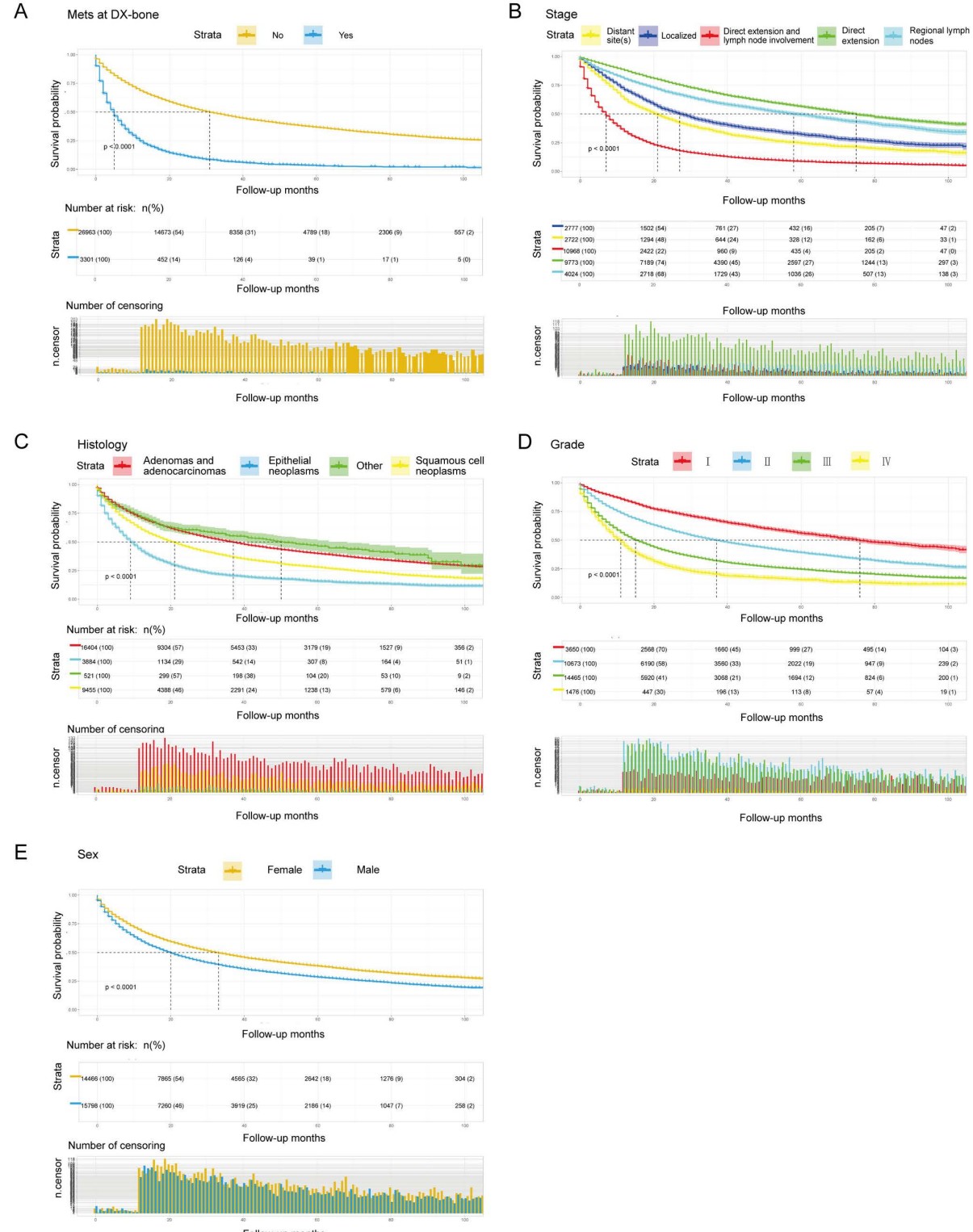

**Fig 3. Kaplan-Meier curves for OS analysis of the primary upper lobe lung cancer patients stratified by (A) Mets at DX-bone; (B) Stage; (C) Histology; (D) Grade and (E) Sex.**

**Table 2. Univariate and multivariate Cox regression model analysis of the lung cancer patients with primary site in the upper lobe.**

| Variables | Univariable | | Multivariable | |
|---|---|---|---|---|
| | HR (95%CI) | P value | HR (95%CI) | P value |
| **Factors Selected** | | | | |
| **Age** | | | | |
| 20-54 | 1[Reference] | NA | 1[Reference] | NA |
| 55-74 | 0.745(0.724-0.767) | <0.001*** | 1.236(1.171-1.305) | <0.001*** |
| ≥75 | 1.488(1.445-1.533) | <0.001*** | 1.807(1.707-1.913) | <0.001*** |
| **Race** | | | | |
| White | 0.817(0.775-0.862) | <0.001*** | NA | NA |
| Black | 1.106(1.059-1.155) | <0.001*** | 1.056(1.012-1.103) | 0.013* |
| Other | 1[Reference] | NA | 1[Reference] | NA |
| **Sex** | | | | |
| Male | 1.317 (1.280-1.335) | <0.001*** | 1.208(1.174-1.244) | <0.001*** |
| Female | 1[Reference] | NA | 1[Reference] | NA |
| **Grade** | | | | |
| I | 0.451(0.428-0.476) | <0.001*** | 0.610(0.560-0.665) | <0.001*** |
| II | 0.723(0.702-0.746) | <0.001*** | 0.822(0.765-0.883) | <0.001*** |
| III | 1.669(1.622-1.717) | <0.001*** | 0.973(0.911-1.039) | 0.410 |
| IV | 1[Reference] | NA | 1[Reference] | NA |
| **Histology type** | | | | |
| Epithelial neoplasms | 1.987(1.912-2.064) | <0.001*** | 1.232(1.083-1.401) | 0.002** |
| Squamous cell neoplasms | 1.211(1.175-1.248) | <0.001*** | 1.109(0.980-1.256) | 0.102 |
| Adenomas and adenocarcinomas | 0.639(0.621-0.658) | <0.001*** | 0.915(0.810-1.034) | 0.156 |
| Other | 1[Reference] | NA | 1[Reference] | NA |
| **Stage** | | | | |
| Regional lymph nodes involved only | 0.930(0.886-0.977) | 0.004** | 1.866(1.763-1.976) | <0.001*** |
| Regional by direct extension only | 0.569(0.543-0.596) | <0.001*** | 1.355(1.283-1.430) | <0.001*** |
| Regional by both direct extension and lymph node involved | 1.169(1.115-1.225) | <0.001*** | 2.129(2.014-2.251) | <0.001*** |
| Distant site(s) involved | 3.868(3.756-3.982) | <0.001*** | 3.222(3.086-3.364) | <0.001*** |
| Localized only | 1[Reference] | NA | 1[Reference] | NA |
| **Regional lymph nodes removed** | | | | |
| Yes | 0.285(0.276-0.294) | <0.001*** | 0.470(0.453-0.486) | <0.001*** |
| No | 1[Reference] | NA | 1[Reference] | NA |
| **Mets at DX -Bone** | | | | |
| Yes | 3.601(3.461-3.747) | <0.001*** | 1.404(1.344-1.468) | <0.001*** |
| No | 1[Reference] | NA | 1[Reference] | NA |
| **Mets at DX-Liver** | | | | |
| Yes | 4.076(3.862-4.301) | <0.001*** | 1.545(1.459-1.635) | <0.001*** |
| No | 1[Reference] | NA | 1[Reference] | NA |

HR, hazard ratio; CI, confidence interval; NA, not applicable; *P<0.05, **P<0.01, ***P<0.001.

| Subgroups | No. of Patients | No. of Patients with events | p-Value | | Hazard Ratio(95%CI) |
|---|---|---|---|---|---|
| **Age** | | | | | |
| 18-54 | 2588 | 1494 | | | 1(Reference) |
| 55-74 | 18016 | 10614 | <0.001*** | | 1.236(1.171-1.305) |
| ≧75 | 9691 | 7124 | <0.001*** | | 1.807(1.707-1.913) |
| **Race** | | | | | |
| White | 24213 | 15361 | | | |
| Black | 3545 | 2358 | 0.013* | | 1.056(1.012-1.103) |
| Other | 2537 | 1513 | | | 1(Reference) |
| **Sex** | | | | | |
| Male | 15818 | 10764 | <0.001*** | | 1.208(1.174-1.244) |
| Female | 14477 | 8468 | | | 1(Reference) |
| **Grade** | | | | | |
| I | 3655 | 1502 | <0.001*** | | 0.610(0.560-0.665) |
| II | 10684 | 6100 | <0.001*** | | 0.822(0.765-0.883) |
| III | 14478 | 10425 | 0.410 | | 0.973(0.911-1.039) |
| IV | 1478 | 1205 | | | 1(Reference) |
| **Histology type** | | | | | |
| Epithelial neoplasms | 3887 | 3176 | 0.002** | | 1.232(1.083-1.401) |
| Squamous cell neoplasms | 9467 | 6471 | 0.102 | | 1.109(0.980-1.256) |
| Adenomas and adenocarcinomas | 16419 | 9318 | 0.156 | | 0.915(0.810-1.034) |
| Other | 522 | 267 | | | 1(Reference) |
| **Stage** | | | | | |
| Regional lymph nodes involved only | 2779 | 1753 | <0.001*** | | 1.866(1.763-1.976) |
| Regional by direct extension only | 4024 | 1931 | <0.001*** | | 1.355(1.283-1.430) |
| Regional by both direct extension and lymph node involvement | 2723 | 9649 | <0.001*** | | 2.129(2.014-2.251) |
| Distant site(s) involved | 10980 | 3924 | <0.001*** | | 3.222(3.086-3.364) |
| Localized only | 9789 | 1975 | | | 1(Reference) |
| **Regional lymph nodes removed** | | | | | |
| Yes | 13509 | 5682 | <0.001*** | | 0.470(0.453-0.486) |
| No | 16786 | 13550 | | | 1(Reference) |
| **Mets at diagnosis -Bone** | | | | | |
| Yes | 3302 | 3077 | <0.001*** | | 1.404(1.344-1.468) |
| No | 26993 | 16155 | | | 1(Reference) |
| **Mets at diagnosis -Liver** | | | | | |
| Yes | 1574 | 1502 | <0.001*** | | 1.545(1.459-1.635) |
| No | 28721 | 17730 | | | 1(Reference) |

**Fig 4. Forest plot for the hazard ratio analysis of all-cause death in the primary upper lobe lung cancer patients. CI, confidence interval; *P<0.05, **P<0.01, ***P<0.001.**

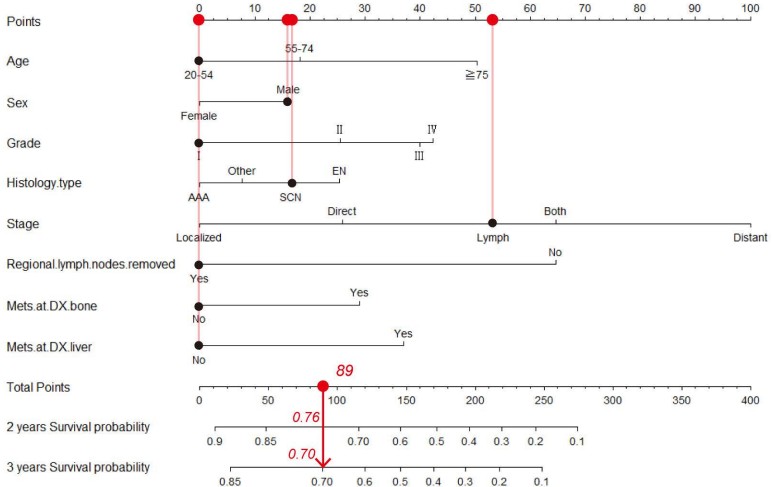

**Fig 5. Prognostic nomogram for predicting the 2-year and 3-year OS in the primary upper lobe lung cancer patients. AAA, adenomas and adenocarcinomas; SCN, squamous cell neoplasms; EN, epithelial neoplasms; the concordance index= 0.761.**

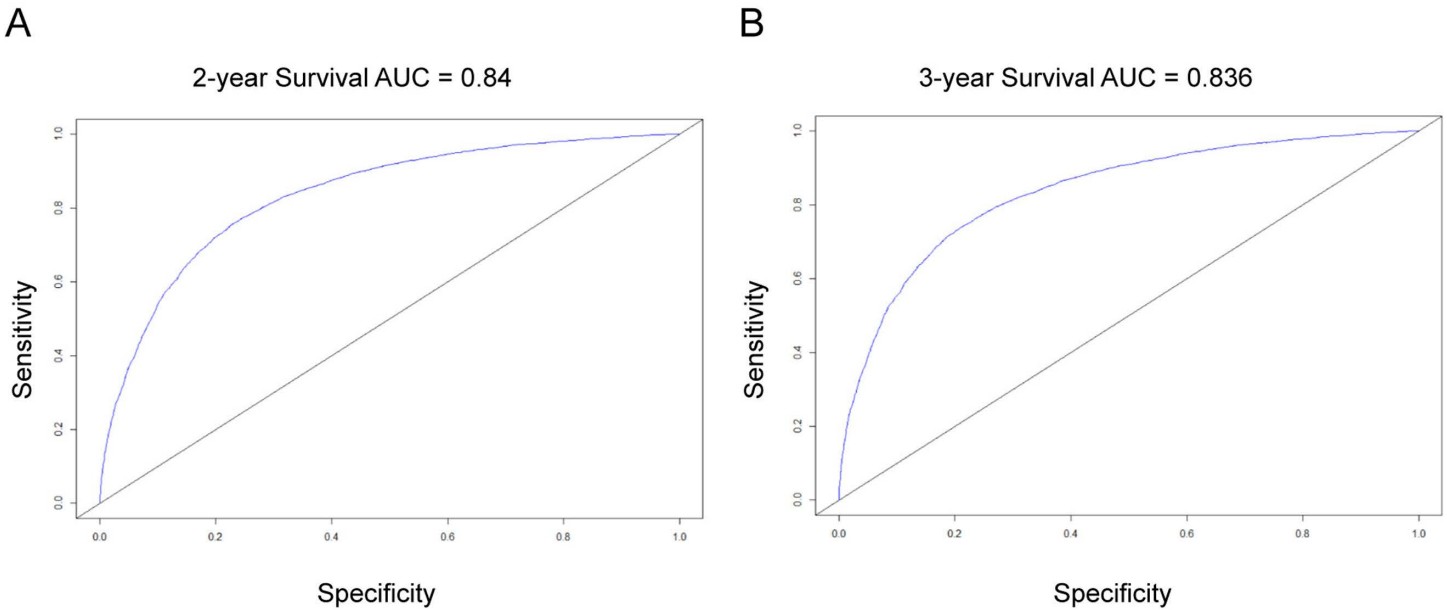

**Fig 6. Time-dependent ROC curve for predicting the 2-year and 3-year OS probability in the primary upper lobe lung cancer patients. A, the 2-year survival AUC; B, the 3-year survival AUC. ROC, receiver operating characteristic; AUC, area under the ROC curve.**

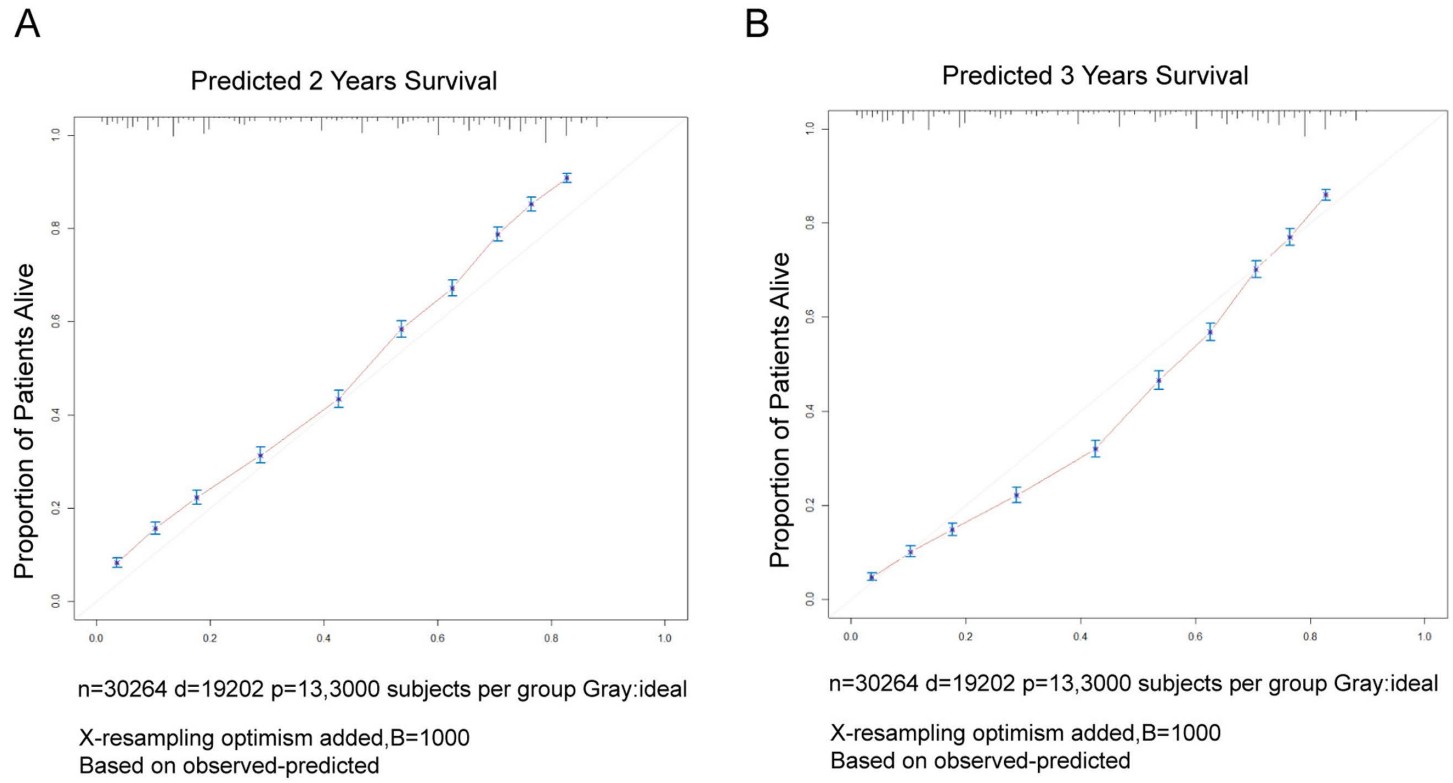

**Fig 7. Calibration plot for evaluating the predicted 2- and 3-year survival and the actual outcome in the primary upper lobe lung cancer patients.**

(B) OS in patients with primary upper lobe lung cancer. The light red line represents the ideal reference line, where the predicted survival probability perfectly matches the observed survival rate. The red dots, calculated using the bootstrapping method (sample size: 1000), indicate the performance of the nomogram. The closer the solid red line is to the light red reference line, the more accurate the model's predicted survival rate. As shown in Fig 7, the nomogram's calibration curve demonstrates a high degree of consistency between predicted and observed survival rates, indicating excellent discriminative and calibration capabilities of the model. In conclusion, the nomogram for patients with primary upper lobe lung cancer exhibits high accuracy and reliability in predicting 2-year and 3-year survival rates.

## Discussion

Lung cancer remains the leading cause of cancer-related mortality worldwide, primarily classified into two histological types: small-cell lung cancer (SCLC) and non-small-cell lung cancer (NSCLC). Due to the majority of patients being diagnosed at advanced stages with poor prognosis, prognostic research on lung cancer has consistently been a focal point in clinical practice. At first, we compared the number of lung cancer patients with different primary tumor distributions across the upper, middle and lower lobes, and revealing that the upper lobe was the most prevalent location, likely due to its anatomical susceptibility to airborne carcinogens (e.g., smoking-related particles) [12]. Previous studies have also observed that lung cancer occur more frequently in the upper lobes compared to the middle and lower lobes [9,13,14],which aligns with our findings. Notably, our survival curve analysis demonstrated that patients with primary lung cancer in the upper lobes exhibited lower survival probabilities compared to those with primary sites in the middle lobes, while no statistical difference was observed between upper and lower lobes. This suggests that the upper lobe location may serve as a significant factor influencing clinical outcomes.Indeed, factors related to progression and prognosis of primary lung cancer in different lobes worth further exploration.

Current research on the association between primary tumor location and prognosis in lung cancer remains controversial, particularly regarding survival differences and underlying mechanisms between upper lobe tumors and those in the lower/middle lobes. Several studies support a prognostic advantage for upper lobe tumors. For instance, a meta-analysis by Lee et al. [15]demonstrated that among stage I-III NSCLC patients, the 5-year survival rate was significantly higher for upper lobe tumors compared to non-upper lobe tumors (middle + lower lobes), while no significant survival differences were observed between lower vs. non-lower lobes or upper vs. middle/lower lobes. Takamori et al. [16]further reported that upper lobe tumors treated with programmed cell death-1 (anti-PD-1) therapy exhibited superior progression-free survival (PFS) and OS compared to non-upper lobe tumors. This discrepancy may be aPFSttributed to the higher tumor mutational burden (TMB) observed in upper/middle lobe squamous cell carcinomas (SCCs), as TMB is a critical predictor of immunotherapy efficacy. These findings appear to indicate a favorable prognosis for upper lobe tumors, which contradicts the results of our study.

This discrepancy may stem from heterogeneity in study populations and methodologies. Previous studies predominantly focused on single-stage cohorts (e.g., exclusively stage I-III or IV patients), whereas the current study encompassed an all-stage (I-IV) population. The increased proportion of advanced-stage cases may amplify the metastatic propensity of upper lobe tumors, thereby counteracting the survival benefits from early-stage surgical resection. Moreover, while existing literature often merges the middle lobe into the "non-upper lobe" group for analysis, our study evaluated the middle lobe separately. Such classification discrepancies could lead to biased outcomes.

The association between tumor location and prognosis in lung cancer can be explained through multiple mechanisms. Firstly, there are significant differences in lymph node metastasis patterns: upper lobe tumors exhibit skip metastasis (i.e., direct metastasis to mediastinal N2 lymph nodes bypassing N1 nodes), leading to occult metastasis that complicates staging and therapeutic efficacy[17]. In contrast, middle lobe tumors, due to their limited lymphatic drainage, are more amenable to complete surgical resection[18]. Secondly, heterogeneity in gene expression likely drives prognostic disparities. Epidermal Growth Factor Receptor (EGFR) mutations were more prevalent in upper lobe tumors, which theoretically benefit from targeted therapy[19,20]. Middle lobe tumors showed a higher PD-L1 expression[21], rendering them sensitive to anti-PD-1. In contrast, lower lobe tumors exhibit a higher frequency of Anaplastic Lymphoma Kinase gene (ALK) rearrangements (52% vs. 34% vs. 36%, *p<0.05*). Compared to EGFR+ or EGFR−/ALK− tumors, ALK+ tumors are more strongly associated with the absence of pulmonary metastasis and the presence of lymphangitic carcinomatosis, distant lymph node metastasis, and sclerotic bone metastasis[21].

Anatomical characteristics and therapeutic challenges significantly impact prognosis. Although upper lobe tumors offer a clearer surgical field, their proximity to the subclavian vessels and mediastinal structures, coupled with a high rate of occult mediastinal metastasis [22], complicates resection. Conversely, lower lobe tumors, adjacent to the diaphragm and esophagus, are prone to pleural or intra-abdominal organ invasion, posing greater surgical difficulty. Additionally, their frequent comorbidity with interstitial pulmonary disease (IPF) elevates the postoperative risk of acute exacerbation by 30% [23]. Furthermore, the anatomical dependency of metastatic patterns may exacerbate survival disparities: Shan et al.[24]reported that in stage IV NSCLC, upper lobe primaries were more likely to metastasize to the brain, middle lobe tumors predominantly spread intrapulmonarily, and lower lobe tumors favored bone metastasis.In conclusion, the prognostic impact of tumor location arises from the complex interplay of anatomical constraints, molecular heterogeneity, and therapeutic responses. Our findings emphasize that relying solely on lobar classification is insufficient to predict survival. Instead, clinical strategies should integrate driver gene profiles, metastatic patterns, and comorbidity risks to optimize personalized management.

Although previous studies have proposed various nomogram models for lung cancer prognosis, existing reports based on the SEER database have rarely focused on the impact of primary lung cancer sites, with even fewer conducting in-depth analyses[25–27]. This study represents the first to specifically investigate primary tumor locations in lung cancer. Through systematic analysis of the SEER cohort, we developed a nomogram for predicting OS in patients with primary upper lobe lung cancer, aiming to provide crucial references for clinical decision-making. After screening and analyzing key prognostic factors, the study identified eight independent factors associated with primary upper lobe lung cancer: age, gender, tumor grade, histological type, stage, regional lymph node removal, Mets at DX-bone and Mets at DX-Liver. The constructed nomogram enables personalized prediction of 1-year and 3-year OS, while quantitatively demonstrating the relative contributions of each factor to prognosis.

The developed nomogram demonstrated excellent performance in predicting OS in patients with primary upper lobe lung cancer. First, the model exhibited a high discriminative ability with a C-index value of 0.761, effectively distinguishing patient groups with different mortality risks. Second, the diagnostic accuracy of the model was further validated through ROC curve analysis, with AUC values of 0.840 and 0.836 for 2-year and 3-year OS predictions, respectively, indicating high accuracy in identifying patient survival outcomes.

Age is a critical prognostic factor for primary upper lobe lung cancer[28–30], and its impact aligns with our general understanding of the relationship between age and disease prognosis. This study found that the age group of 70–74 years is a high-risk period for

lung cancer incidence, with age showing a positive correlation with mortality risk and poor prognosis. The biological characteristics of elderly patients, such as declining physiological functions, higher risks of comorbidities, and reduced tolerance to treatment, may complicate therapy and affect survival, leading to worse outcomes[31]. Therefore, treatment decisions for elderly patients in clinical practice require more nuanced evaluation to ensure the safety and effectiveness of treatment plans[32].

This study revealed that female patients with upper lobe lung cancer exhibited significantly better survival prognosis compared to males, a disparity potentially mediated by multifactorial synergies[33]. Firstly, distinct clinicopathological and behavioral patterns were observed: females demonstrated a higher proportion of adenocarcinomas and more early-stage cases at diagnosis, while males exhibited greater smoking exposure rates that may contribute to more aggressive tumor phenotypes. Secondly,at the molecular level, EGFR mutations mostly occur in adenocarcinoma, younger women and girls, and never-smokers, whereas KRAS mutations showed higher prevalence in male smokers of non-Asian ethnicity[34–37]. Notably, multiple phase III randomized controlled trials (RCTs) [38–40]have established EGFR tyrosine kinase inhibitors (TKIs) as the first-line therapy for EGFR-mutant NSCLC, demonstrating superior progression-free survival, objective response rates, and quality of life compared to conventional chemotherapy. In contrast, development of therapeutics to target KRAS-mutant phenotype has been remarkably frustrating[41].

Additionally, estrogen has been recognized as a promoting factor in the initiation and progression of lung cancer, which appears contradictory to our findings[35,42]. This discrepancy may be attributed to the inclusion of postmenopausal women in this study. Regarding the impact of gender on the efficacy of immunotherapy for non-small cell lung cancer (NSCLC), current studies have yet to reach a consensus. The research by Conforti et al.[43]demonstrated that compared to chemotherapy alone, anti-PD-1/PD-L1 therapy combined with chemotherapy could provide more significant OS benefits for female patients. However, the Valencia team [44] proposed that the key determinant of immune checkpoint inhibitor efficacy is not gender per se, but rather the 17β-estradiol/ERα/PD-L1 signaling circuit present in the tumor microenvironment. Future research should focus on establishing large-scale clinical databases incorporating multiple dimensions such as menstrual status, hormonal levels, and treatment sequencing, combined with multi-omics data analysis to elucidate gender-specific biological mechanisms, thereby guiding personalized therapeutic approaches.

Regional lymph node dissection and pathologic examination could offer accurate clinical staging and prognostic information, as well as improve cure rates and have survival benefits[45]. Concurrently, thorough dissection eliminates micrometastases, reduces local recurrence risks, and improves patient survival[46]. However, the extent of dissection requires careful balancing between staging accuracy and surgical safety. Our findings underscore the importance of standardized lymph node dissection and suggest that it should be implemented as a critical component of lung cancer surgery in clinical practice to optimize patient outcomes[47].

Mets at DX-bone and Mets at DX-Liver are independent risk factors influencing the prognosis of lung cancer.According to the historical records, bone metastases accounted for 30–40% of the lung cancer patients, and unfortunately, these patients had to experience skeletal complications, such as cancer-induced bone pain, hypercalcemia, pathological bone fractures, and cancer cachexia[48], leading to severe impairment of their quality of life. Liver metastases are common in patients with metastases from small cell lung cancer, and these patients have poor survival rate[49].

The ninth edition of the TNM staging system by the International Association for the Study of Lung Cancer (IASLC) reaffirms the core prognostic value of tumor grade and stage,

yet fails to incorporate critical factors such as histopathological subtypes[50].Through multivariable analysis, this study identified that in addition to the overall TNM stage (I-IV), age, sex, histology type, regional lymph nodes removed, and Mets at DX-bone/liver are independent prognostic factors for upper lobe lung cancer. By integrating TNM staging with histological classification, our model addresses limitations in the current staging system.

Under the precision stratified treatment framework for lung cancer (early-stage surgery combined with adjuvant therapy, locally advanced disease with immunoconsolidation, advanced-stage targeted/immunotherapy dominance)[36,37], the upper-lobe lung cancer nomogram developed in this study (C-index=0.761) demonstrates multidimensional clinical-translational value:The model can identify high-risk patients, prompting physicians to optimize surgical decision-making.Furthermore, in terms of adjuvant therapy, the high-risk group may require postoperative combination of targeted therapy and immunotherapy, while the low-risk group can undergo de-escalated treatment. Additionally, dynamic prognostic monitoring can be achieved by integrating treatment response data (such as postoperative ctDNA clearance[51]), allowing the model to be upgraded in the future to a dynamic one and realizing closed-loop management of "treatment-prognosis". Although the current model relies on clinical variables, it is superior to traditional staging.

This study also has several limitations. Firstly, as a retrospective study, these results have to consider the inherent selection biases.Secondly, the SEER database lacks some key factors, such as surgical method,types of raiation therapy and information on chemotherapy, immunotherapy and metastasis sites.These unmeasured variables may cause confounding bias, which could affect the findings. Thirdly, although our nomogram was internally validated using bootstrapping validation, other models proposed by external validation are required for future exploration.More importantly, relying solely on clinical variables may inadequately reflect the molecular heterogeneity of lung cancer.In the current era of precision medicine, driver gene status (e.g., EGFR mutations, ALK rearrangements) directly influences responses to targeted therapies, while PD-L1 expression levels serve as crucial predictive biomarkers for immune checkpoint inhibitor efficacy. These molecular features have been incorporated into the prognostic stratification system of NCCN guidelines[52].However, the absence of such molecular information in SEER data limits this study. Future integration of molecular biomarkers could guide systemic therapy for molecularly defined subgroups of advanced disease patients, significantly enhancing the clinical utility of predictive models.

## Conclusion

In summary, we established and validated a nomogram for predicting 2-year and 3-year survival probability of lung cancer patients with primary site in the upper lobe, which performed well in discrimination and calibration. This novel nomogram might serve as an important early warning tool in favor of individualized clinical therapeutic regimen development for lung cancer patients.

## Author contributions

**Data curation:** Qizhuo Hou.

**Formal analysis:** Wenze Yu.

**Funding acquisition:** Lu Long, Bin Yi.

**Methodology:** Wenze Yu.

**Software:** Wenze Yu, Qizhuo Hou.

**Visualization:** Wenze Yu.

**Writing – original draft:** Wenze Yu.

**Writing – review & editing:** Bin Yi.

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
