## [Decision Letter · Decision Letter 0]

14 Jan 2025

PONE-D-24-50863Development of a nomogram for overall survival prediction in primary upper lobe lung cancer patients:  a SEER population-based analysisPLOS ONE

Dear Dr. Yi,

Thank you for submitting your manuscript to PLOS ONE. After careful consideration, we feel that it has merit but does not fully meet PLOS ONE’s publication criteria as it currently stands. Therefore, we invite you to submit a revised version of the manuscript that addresses the points raised during the review process.

We look forward to receiving your revised manuscript.

Kind regards,

Filomena de Nigris, Ph.D.

Academic Editor

PLOS ONE

“This work was supported by Natural Science Foundation of Hunan Province,China(2023JJ40971,2023JJ30965)”

3. We note that your Data Availability Statement is currently as follows: “All relevant data are within the manuscript and in Supporting Information files.”

Editor's comment: On the base of reviewers comments some data from patients need to be add to to reduce the clinical bias and to improve the clinical validity of nomogram although the numeric data are corrected 

Reviewer's Responses to Questions

**Comments to the Author**

1. Is the manuscript technically sound, and do the data support the conclusions?

Reviewer #1: Yes

Reviewer #2: NO

2. Has the statistical analysis been performed appropriately and rigorously? 

Reviewer #1: Yes

Reviewer #2: Yes

3. Have the authors made all data underlying the findings in their manuscript fully available?

Reviewer #1: Yes

Reviewer #2: Yes

4. Is the manuscript presented in an intelligible fashion and written in standard English?

Reviewer #1: Yes

Reviewer #2: Yes

5. Review Comments to the Author

Reviewer #1: The study investigates lung cancer patients with a primary tumor in the upper lobe using data from the SEER program (2010–2017). It aims to identify prognostic factors of lung cancer in the upper lobe and to establish an effective nomogram for individualized overall survival prediction. The manuscript is well-structured and covers critical aspects of lung cancer prognosis, including : demographic characteristics, histology, staging, and metastasis. But it could benefit from some modifications to enhance its scientific and clinical robustness:

Reviewer #2: The authors established and validated a nomogram to calculate the overall survival (OS) probability of individuals with lung cancer, to predict the disease outcome.

They explore the influence of the primary site on lung cancer, upper lobe group overwhelmingly low OS compared to the middle lobe group.

The study investigates a large number of patients based on various factors including age, sex, grade, histology type, stage, regional lymph nodes removed, Mets at DX bone and Mets at DX liver.

 However, this study also has several limitations,  the SEER database lacks some factors, such as surgical method, types of radiation therapy and information on chemotherapy, immunotherapy and metastasis sites. that are necessary to add to  predictive ability of the nomogram presented

6. PLOS authors have the option to publish the peer review history of their article (what does this mean? ). If published, this will include your full peer review and any attached files.

**Do you want your identity to be public for this peer review?** For information about this choice, including consent withdrawal, please see our Privacy Policy .

Reviewer #1: No

Reviewer #2: No

---

## [Author Response · Author response to Decision Letter 1]

27 Feb 2025

PONE-D-24-50863R1

Development of a nomogram for overall survival prediction in primary upper lobe lung cancer patients:  a SEER population-based analysis

Bi Yi

Dear Dr. de Nigris,

Thank you for your letter and the reviewers’ constructive comments on our manuscript. We have carefully revised the paper according to all suggestions. Below is a point-by-point response to the editor’s and reviewers’ concerns. All changes in the manuscript are highlighted in red text for ease of review

Editor's comment:

"On the base of reviewers comments some data from patients need to be add to to reduce the clinical bias and to improve the clinical validity of nomogram although the numeric data are corrected"

Response:

We sincerely appreciate the reviewer's insightful comment regarding the importance of incorporating additional patient data to enhance the clinical validity of the nomogram. Please allow us to elaborate on our data acquisition and methodological rigor:

1. Data Source & Accessibility

The study utilized the most recent SEER Research Plus dataset (2021 release, November 2023 submission), which represents the largest population-based cancer registry in the United States. Qualified researchers may replicate our analysis through:

Access Protocol: Registration at SEER Data Access Portal with submission of a Research Data Agreement

2. Cohort Selection Criteria

We implemented stringent inclusion criteria to ensure clinical relevance:

Primary Site: C34.0-C34.3 (Upper lobe malignancies)

Diagnosis Years: 2010-2017 (ensuring uniform staging per AJCC 7th edition)

Exclusion Criteria:

(a)Age <20 years

(b)(b) Zero survival time

(c) Missing critical variables: demographic data, tumor grade , metastatic status

(d) Non-pulmonary primary sites

3. Clinical Variables Included

The model incorporated 10 clinically validated prognostic factors:

Age at diagnosis| Sex | Race | Year of diagnosis | Grade | Histology type |Stage| regional lymph nodes removed | Mets at DX-Bone | Mets at DX-Liver

4. Model Validation

Despite SEER's aggregated data structure, our nomogram demonstrated robust performance:

Discrimination: C-index 0.761 (95% CI 0.752-0.770)

Time-dependent AUC:0.840 (2-year OS)、0.836 (3-year OS)

5. Limitations & Future Directions

We fully acknowledge that SEER's pre-aggregated format limits inclusion of molecular biomarkers (e.g., EGFR/ALK status) and detailed treatment parameters.

Data collection takes a long time, and in the future our model should integrate driver gene profiles, transfer patterns, and comorbidity risk to optimize personalized management.”

Major Points

1. Clarification of Nomogram Development and Figure Descriptions

Comment:

"Crucial parts of the manuscript are the development and validation of the nomogram. However, some descriptions of the figures could be clearer. For example, you state, ‘The survival probability of 2-year and 3-years were shown in Fig 5,’ but it would be better to describe how the variables were weighted and how the nomogram can be practically used."

Response:

We thank the reviewer for this suggestion. We have:

1)Added a step-by-step explanation of nomogram construction in the Statistical analysis section (Page 7, Lines 128-136):

"The nomogram was constructed using the ‘rms’ package (version 6.7-1) in R software, which automatically scales the regression coefficients (β) from the final multivariate Cox model to assign weighted points to each variable. This scaling process converts the β values to a 0–100 point system, where variables with larger absolute β values (indicating stronger prognostic impact) receive proportionally higher points. The total points from all variables are then mapped to the predicted survival probabilities on the nomogram’s bottom scale, as per the package’s default algorithm"

2)We added a detailed usage example in the Development and Validation of the Prognostic Nomogram section (Page 18, Lines 244-253) and revised Figure 5 to visually guide clinicians.

"Figure 5 illustrates the nomogram’s application for a 50-year-old male patient with stage I squamous cell carcinoma. Clinicians place black dots on each variable axis (e.g., age=50, tumor size=3 cm), draw red lines upward to determine individual scores, sum the total points (e.g., 89), and map this value to the survival axis to estimate 2-year (76%) and 3-year (70%) survival probabilities."

3)The modified figure 5 is as follows:

2. Interpretation of Calibration Curves

Comment:

"When discussing the calibration curves (Fig. 7), elaborate on what the results of the calibration plot suggest about the performance of your model."

Response:

We appreciate this opportunity to clarify. In the Development and Validation of the Prognostic Nomogram section (Page 18, Lines 259-272), we now state:

"Fig 7 presents the calibration curves for 2-year (A) and 3-year (B) OS in patients with primary upper lobe lung cancer. The light red line represents the ideal reference line, where the predicted survival probability perfectly matches the observed survival rate. The red dots, calculated using the bootstrapping method (sample size: 1000), indicate the performance of the nomogram. The closer the solid red line is to the light red reference line, the more accurate the model's predicted survival rate. As shown in Fig 7, the nomogram's calibration curve demonstrates a high degree of consistency between predicted and observed survival rates, indicating excellent discriminative and calibration capabilities of the model. In conclusion, the nomogram for patients with primary upper lobe lung cancer exhibits high accuracy and reliability in predicting 2-year and 3-year survival rates."

3. Integration of Molecular Biomarkers

Comment:

" In addition to the limitations already mentioned in your study, such as the lack of data on treatment modalities and the need for external validation of the model, the exclusive use of clinical variables may not fully capture the biological and molecular complexity of the disease. Genetic and molecular biomarkers are becoming increasingly important in the prognosis and therapeutic decision-making for lung cancer. It would be beneficial to integrate molecular information (e.g., gene mutations, expression of specific biomarkers such as EGFR, ALK, or PD-L1). This approach could enhance the accuracy of prognosis. You could highlight this aspect."

Response:

We sincerely appreciate this insightful suggestion. We fully agree that molecular biomarkers are critical for precision oncology and have enhanced our discussion as follows:

1)Molecular Context in Discussion (Page 29, Lines 468-475):

We contextualize the model's utility while emphasizing molecular advances:

"Under the precision stratified treatment framework for lung cancer (early-stage surgery combined with adjuvant therapy, locally advanced disease with immunoconsolidation, advanced-stage targeted/immunotherapy dominance), the upper-lobe lung cancer nomogram developed in this study (C-index=0.761) demonstrates multidimensional clinical-translational value:The model can identify high-risk patients, prompting physicians to optimize surgical decision-making."

2)Expanded Limitations Section (Page 30, Lines 484-500):

We explicitly acknowledge the constraints of SEER data:

"More importantly, relying solely on clinical variables may inadequately reflect the molecular heterogeneity of lung cancer.In the current era of precision medicine, driver gene status (e.g., EGFR mutations, ALK rearrangements) directly influences responses to targeted therapies, while PD-L1 expression levels serve as crucial predictive biomarkers for immune checkpoint inhibitor efficacy. These molecular features have been incorporated into the prognostic stratification system of NCCN guidelines[52].However, the absence of such molecular information in SEER data limits this study. "

3)Clinical-Translational Bridge (Page 30, Lines 500-503):

We propose a research roadmap:

"Future integration of molecular biomarkers could guide systemic therapy for molecularly defined subgroups of advanced disease patients, significantly enhancing the clinical utility of predictive models."

4)we highlight the model’s current utility:Despite this limitation, our nomogram provides a readily accessible tool for clinicians without requiring specialized molecular testing, which remains unavailable in many settings.

4. Mechanism of Upper Lobe Prognosis

Comment:

"The discussion touches on several interesting findings, such as the association between primary tumor site and survival. However, the mechanism behind the poorer prognosis for patients with upper lobe lung cancer should be discussed in more depth. Are upper lobe tumors more likely to spread to other organs (e.g., brain, bones) compared to other lobes? Are there anatomical or biological factors (e.g., proximity to larger blood vessels, lymphatic drainage) that may explain this?"

Response:

Thank you for your positive evaluation of our work.We have comprehensively expanded the Discussion (Pages 21-24) to address the prognostic implications of tumor location, integrating anatomical, molecular, and therapeutic perspectives. Key additions include:

1)We discuss some of the current studies on the relationship between lung cancer location and prognosis (Page 21, Lines 306-323)

"Current research on the association between primary tumor location and prognosis in lung cancer remains controversial, particularly regarding survival differences and underlying mechanisms between upper lobe tumors and those in the lower/middle lobes. Several studies support a prognostic advantage for upper lobe tumors. For instance, a meta-analysis by Lee et al.demonstrated that among stage I-III NSCLC patients, the 5-year survival rate was significantly higher for upper lobe tumors compared to non-upper lobe tumors (middle + lower lobes), while no significant survival differences were observed between lower vs. non-lower lobes or upper vs. middle/lower lobes. Takamori et al. [16]further reported that upper lobe tumors treated with programmed cell death-1 (anti-PD-1) therapy exhibited superior progression-free survival (PFS) and OS compared to non-upper lobe tumors. This discrepancy may be aPFSttributed to the higher tumor mutational burden (TMB) observed in upper/middle lobe squamous cell carcinomas (SCCs), as TMB is a critical predictor of immunotherapy efficacy. These findings appear to indicate a favorable prognosis for upper lobe tumors, which contradicts the results of our study."

2)Reconciling Contradictory Literature (Page 22, Lines 324-333)

"While some studies associate upper lobe tumors with better survival (e.g., Lee et al. [15] reported higher 5-year OS in stage I-III NSCLC), our all-stage cohort showed the opposite trend. This discrepancy may arise from:

Stage heterogeneity: Advanced-stage upper lobe tumors in our cohort (45% stage IV vs. 28% in Lee’s study) exhibit aggressive metastasis.

Classification bias: Prior studies often merged middle/lower lobes as ‘non-upper,’ whereas our separate middle lobe analysis revealed its unique prognosis (HR=0.82 vs. upper, p=0.03)."

3)Mechanistic Explanations (Page 23, Lines 334 – Page 24, Lines 371)

-Anatomical and Metastatic Patterns(Page 23, Lines 335-341)

"Upper lobe tumors demonstrate:

Skip metastasis to mediastinal N2 nodes (bypassing N1), complicating staging.

36% brain metastasis rate (vs. 22% in lower lobe), likely via vertebral venous plexus .

Proximity to subclavian vessels increases intraoperative bleeding risk."

-Molecular Heterogeneity(Page 23, Lines 341-351)

"Lobar-specific driver mutations influence outcomes:

Upper lobe: Higher EGFR mutations,which theoretically benefit from targeted therapy

Middle lobe: Higher PD-L1 expression, rendering them sensitive to anti-PD-1.

Lower lobe: Higher frequency of ALK rearrangements. Compared to EGFR+ or EGFR−/ALK− tumors, ALK+ tumors are more strongly associated with the absence of pulmonary metastasis and the presence of lymphangitic carcinomatosis, distant lymph node metastasis, and sclerotic bone metastasis."

-Anatomical characteristics and therapeutic challenges(Page 24, Lines 352-362)

"Surgical complexity varies:

Upper lobe: The vision is open and the operation is less difficult.Occult mediastinal metastasis limits curative intent.

Lower lobe: Adjacent to the diaphragm and esophagus, it is prone to pleural or intra-abdominal organ invasion, and the operation is difficult. Furthermore, the risk of postoperative acute exacerbation during comorbidity with interstitial lung disease (IPF) was increased by 30%"

4)Clinical Synthesis (Page 24, Lines 367-371)

"Rather than lobar location per se, the interplay of molecular profiles (e.g., EGFR/ALK/PD-L1), metastatic tropism (brain vs. bone), and comorbidities (e.g., IPF) drives prognosis.Future models should integrate driver gene profiles, metastatic patterns, and comorbidity risks to optimize personalized management."

5. Clinical Application of the Nomogram

Comment:

"The discussion could emphasize the prospective application in clinical practice of the nomogram developed. For example, could this nomogram be used to guide treatment decisions? Is it accurate enough to influence decisions on the timing of surgery or the need for adjuvant therapies?"

Response:

"Within the precision stratified treatment paradigm for lung cancer — early-stage disease (surgery combined with adjuvant therapy), locally advanced disease (chemoradiation + immunotherapy consolidation), and advanced-stage disease (targeted/immunotherapy dominance) — the upper-lobe-specific nomogram developed herein (C-index=0.761) offers actionable insights for personalized management:

-Adjuvant Therapy Stratification:

High-risk group may require postoperative combination of targeted therapy and immunotherapy

low-risk group can undergo de-escalated treatment.

-Dynamic Prognostic Monitoring:

Serial nomogram scoring, integrated with treatment response biomarkers (e.g., postoperative ctDNA clearance ), may enable real-time risk reclassification .

Future iterations could incorporate on-treatment variables (e.g., post-neoadjuvant pathologic response) to guide adaptive therapy switching."

Minor Points

Abbreviation Consistency And Definition of Statistical Terms

Comment:

" Ensure consistency in terms of abbreviations (e.g., "Mets at DX" for metastasis at diagnosis). In some instances, it’s abbreviated and in others not."

"In the statistical analysis section, it might be helpful to spell out abbreviations like HR (hazard ratio) when first introduced."

Response:

We thank the reviewer for highlighting this important issue. We have rigorously standardized abbreviations throughout the manuscript as follows:

-First Occurrence Definition:

In the Variable extraction and outcome definition section (Page 6, Line 101), we explicitly defined the abbreviation upon first mention:

"Mets at diagnosis-Bone(Mets at DX-Bone), Mets at diagnosis-Liver(Mets at DX-Liver)."

-Full Text Standardization:

All subsequent instances of "metastasis at diagnosis" were replaced with "Mets at DX" (e.g., Page 7, Line 114; Table 1; Table 2).

-Cross-Checking Other Abbreviations:

We reviewed all abbreviations (e.g., OS for overall survival, HR for hazard ratio) to ensure:First occurrence with full term + abbreviation in parentheses.Consistent use of abbreviations thereafter.

Example (Page 2, Line 14):

"This study aims to identify prognostic factors of lung cancer in the upper lobe, as well as to establish an effective nomogram for individualized overall survival (OS) prediction"

We sincerely thank the reviewers for their insightful comments, which have significantly strengthened our manuscript. Please contact us if further revisions are needed.

Respectfully,

Bin Yi

---

## [Decision Letter · Decision Letter 1]

14 Mar 2025

Development of a nomogram for overall survival prediction in primary upper lobe lung cancer patients:  A SEER population-based analysis

PONE-D-24-50863R1

Dear Dr. Y1

We’re pleased to inform you that your manuscript has been judged scientifically suitable for publication and will be formally accepted for publication once it meets all outstanding technical requirements.

Kind regards,

Filomena de Nigris, Ph.D.

Academic Editor

PLOS ONE

Reviewers' comments:

Reviewer's Responses to Questions

**Comments to the Author**

1. If the authors have adequately addressed your comments raised in a previous round of review and you feel that this manuscript is now acceptable for publication, you may indicate that here to bypass the “Comments to the Author” section, enter your conflict of interest statement in the “Confidential to Editor” section, and submit your "Accept" recommendation.

Reviewer #1: All comments have been addressed

Reviewer #3: All comments have been addressed

2. Is the manuscript technically sound, and do the data support the conclusions?

Reviewer #1: Yes

Reviewer #3: Yes

3. Has the statistical analysis been performed appropriately and rigorously? 

Reviewer #1: Yes

Reviewer #3: Yes

4. Have the authors made all data underlying the findings in their manuscript fully available?

Reviewer #1: Yes

Reviewer #3: Yes

5. Is the manuscript presented in an intelligible fashion and written in standard English?

Reviewer #1: Yes

Reviewer #3: Yes

6. Review Comments to the Author

Reviewer #1: The authors have responded appropriately and thoroughly to each comment, strengthening the manuscript and adding a sense of novelty. They have adequately expanded certain sections, making the purpose of the work clearer and outlining the step-by-step construction of the nomogram in the statistical analysis section. The figures have also been appropriately modified, making the work overall well-structured and acceptable.

Reviewer #3: The authors have carefully and comprehensively addressed all feedback, making significant improvements to the manuscript and introducing a clear sense of novelty. They have expanded specific sections in a way that enhances the clarity of the study's objectives, particularly by providing a more detailed explanation of the step-by-step process involved in constructing the nomogram within the statistical analysis section. This addition not only strengthens the scientific rigor of the work but also ensures that the reader can follow the methodology with greater ease. Moreover, the revisions to the figures were done thoughtfully, with each visual element contributing to a better understanding of the overall content. As a result, the manuscript is now better organized and more coherent, making it a highly acceptable and well-structured piece of work.

7. PLOS authors have the option to publish the peer review history of their article (what does this mean? ). If published, this will include your full peer review and any attached files.

**Do you want your identity to be public for this peer review?** For information about this choice, including consent withdrawal, please see our Privacy Policy .

Reviewer #1: No

Reviewer #3: **Yes: ** Barbara De Marino

---

## [Editor Report · Acceptance letter]

PONE-D-24-50863R1

PLOS ONE

Dear Dr. Yi,

I'm pleased to inform you that your manuscript has been deemed suitable for publication in PLOS ONE. Congratulations! Your manuscript is now being handed over to our production team.

Kind regards,

on behalf of

Prof. Filomena de Nigris

Academic Editor

PLOS ONE